# Impacts of Prefabrication in the Building Construction Industry

**Patrícia Fernandes Rocha [1],\*, Nuno Oliveira Ferreira [1], Fernando Pimenta [1] and Nelson Bento Pereira [2]** 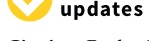

1    Houselab, 4465-097 Matosinhos, Portugal
2    CiCon, 4465-097 Matosinhos, Portugal
\*    Correspondence: patricia@houselab.pt

**Definition:** Interest in sustainable construction has been increasing due to recent events. The limitations of natural resources and the scale of global impacts, specifically as a result of the effects of global climate change, have consequences for the construction sector. These changes are giving rise to a need to reassess the way we face the built environment and rethink new solutions for construction systems or methods that contribute to mitigating negative consequences, among which we highlight the prefabrication method. This new scenario, characterised by the need to meet the decarbonisation goals set for 2050, as well as the effects of the spread of the pandemic crisis, emphasizes the importance of understanding the impacts that may occur in the construction industry, which are essentially understood as increases in sustainability, productivity, quality and, consequently, as reductions in deadlines, costs, and dependence on labour. Therefore, this entry seeks to study on the existing literature on prefabrication, seeking to gather relevant information on the new advances, challenges, and opportunities of this construction method whose approach has been mostly focused on partial or specific aspects for case studies, both highlighting the potential and identifying the gaps and opportunities of prefabrication in this new context. The prefabrication method brings benefits compared to the conventional method, and may be an alternative, as it has more positive global impacts on the environment, the economy, and society, and consequently on the sustainable development of construction, despite some limitations that have been reported and that should be looked into in the future.

**Keywords:** prefabrication; off-site construction; sustainable development; building industry; comprehensive benefits of prefabricated buildings

## 1. Introduction

Recent data show that the construction sector accounts for 32% of global resource consumption and the emission of about 40% of anthropogenic greenhouse gases (GHGs). The waste generated by the sector represents, by mass, about 40% of the materials consumed. In 2019, carbon dioxide emissions, at around 10 $GtCO_2$, reached their highest level, representing 28% of the $CO_2$ emissions associated with energy production worldwide [1,2].

The main targets identified by the European Union (EU) to meet the objectives of the Paris Agreement (Paris Agreement, 2015) include a reduction in the impact of buildings, especially residential buildings. There is a commitment to decarbonise buildings by 2050, setting ambitious targets for tackling climate change. Its realisation in 2030 aims to achieve at least 32% renewable energy in energy consumption and at least a 32.5% improvement in the energy efficiency of buildings in the EU. Based on these data, the International Energy Agency predicts that direct $CO_2$ emissions from buildings should decrease by about 50% and indirect emissions from the sector by 60% in terms of the generation of emissions associated with energy production by 2030. This will result in emissions decreasing by around 6% annually between 2020 and 2030 [3].

However, in 2019, with the spread of the COVID-19 pandemic, another scenario with a relevant global impact in all sectors of activity arose, one of which is the construction industry sector. Despite the changes caused being operational in the first phase, mostly

through a reduction in the production and supply capacity of the markets, resulting in a reduction in global growth, investment, and exports, in the long term, other dynamic scenarios will tend to occur, with some of the consequences beginning to be noted, despite attempts to minimize their impact, mainly on the economy. Among the most mentioned for the construction industry is the acceleration in the integration of digital tools, as they allow for better collaboration, greater control of the value chain, a change to a more data-driven decision-making process (such as the so-called "Digital Twins", which mimic the thermal and energy behaviour of building stock and allow the industry to respond to the problems of the climate crisis), as well as investment in the standardisation of building codes, especially in terms of safety and sustainability, in addition to a greater focus on industrialisation with modularisation, off-site production automation, and on-site assembly automation [3,4].

These recent measures and events, in order to comply with the decarbonisation of buildings so as to mitigate the effects of climate change, will have implications in the construction industry sector, and modifications in the markets are likely to arise with greater constraints on the stabilisation of the skilled workforce and the costs of infrastructure and raw materials, creating the need to develop new digital technologies (the digitisation of products and processes) and new regulations on fundamental matters such as sustainability, with a new emerging dynamic being necessary. In light of these changes, the construction industry sector is beginning to undergo considerable transformations in an ongoing process, transformations which have meanwhile been sped up by the pandemic crisis.

It is therefore pivotal to highlight the role that prefabrication (including modular construction) could play in the medium and long term, since its potential in contributing to an effective reduction in overall building impacts and costs has been discussed. With more environmentally sustainable policies, it will be possible to ensure more energy-efficient and carbon-reduced buildings. The main benefits are the type of use of materials, waste reduction, cost and construction time reduction, the increase in the safety and quality of products, greater efficiency in quality control, the growth of productivity, and the improvement of the performance of buildings [5–8].

One of the main potentials of prefabrication refers to the increase in productivity and the role of construction costs and quality. Essentially, productivity, costs, and the quality of construction are directly affected by their dependence on extensive and complex value chains, by the high number of people involved in these chains, and by labour shortages. One of the potentialities of prefabrication is precisely to allow for an increase in productivity, in cost reduction, and in the rigour and control of the quality of construction.

In fact, many of the obstacles to higher productivity and ways to overcome them have been known for some time, but the industry has been at a standstill. However, there are factors that reduce the obstacles to change: more transparent markets; increased requirements and demands; new technologies and more readily available materials; and also the increasing cost of labour [9]. Those in the construction sector should rethink their operational approaches so as to prevent obsolescence in what could be the next big story of global productivity [10].

The existing literature further shows that the approach to analysing the impacts of the prefabrication method mainly refers to those of an environmental and economic nature.

With regard to the environmental and economic impacts of the construction industry, it is easily observed that poor performance is associated with the intrinsic characteristics of the sector, namely poor quality of buildings, the benefit of price to the detriment of quality, construction errors and defects, the complexity of the value chain, poor integration (of processes, products, and systems), and reduced industrialisation.

Social impacts have not yet been fully analysed. However, there may now be much greater interest as a result of recent changes following the pandemic crisis that has taken place. In the corporate sector, with companies having a social responsibility, the adoption of prefabrication allows for some improvement in the conditions and safety of work, producing effects on the increase in labour productivity [11,12].

This document presents a comprehensive and integrated review of the various scenarios of prefabrication development in the construction industry compared to conventional construction. Moreover, with the aim of outlining the potential of prefabrication in the construction industry in the current context and in light of the goals set forth for 2050 and the consequences of the impact of the spread of the new SARS-CoV virus in this market, the entry will also include an assessment of this conjuncture, seeking guidelines for the future.

One of the strong points of this document is that it aims to aggregate a broad set of bibliographies on this topic, which is somewhat dispersed, identifying gaps and opportunities.

*Background*

Historically, prefabrication in the construction industry had a greater growth with the industrial revolution, in terms of the emergence of new construction solutions and equipment, as well as material processing techniques that allowed for a systematic use of new materials, with large-scale application and series production of standardised prefabricated construction elements.

After the end of World War I, there was a great need for housing, and the best response to this lack was found in the development of serial housing construction, with its rapid assembly and economically acceptable conditions. With the main work being performed in the factory, the output and the economy of means were increased, while the time factor was decreased. This process led to the possibility of developing a large-scale mass production system on a continuous and regular basis over time. For the feasibility of the system, it was necessary to find a standardisation and normalisation of the dimensions of the various components in order to guarantee a standard that would allow for compatibility between the elements and subsystems.

The greatest growth occurred in the post-war period, occurring in Germany from the 1920s–1930s, in the United States during the 1940s and 1950s, and in the United Kingdom, in the 1960s–1970s, and resulted from the need for rapid reconstruction, especially of residential and educational buildings, as a result of the need for construction in short periods of time and at reduced costs. Since the 1970s, architectural firms have begun to show an interest in new building technologies and industrialised construction. In their projects, they incorporate materials and products from the construction industry, seeking flexible systems, as was the example in Wales with the program for the development of school building systems, the School Construction Systems Development (SCSD). However, in this period there were some incidents in the implementation of this type of prefabricated system, such as the collapse of the Ronan Point apartment tower in east London, or the fire in a residence for the elderly built through the Consortium of Local Authorities Special Programme (CLASP) in the United Kingdom, which raised concerns about the safety of prefabricated buildings. Other examples in the context of social housing have also gained negative reputations because they are considered to be of lower quality. Japan was one of the other countries that capitalised on synergies with other manufacturing industries. A high volume of modular units ensured economies of scale and lower production costs, allowing for a greater focus on quality, specifically with regard to earthquake resistance. In the decade between 1950 and 1960, the prefabricated system gave rise to a new approach to the traditional Japanese construction method—the meticulous cutting of wooden parts, which is later complemented by the use of fibre and aluminium panels, fixed to steel frames.

Industrialisation therefore radically transformed construction, since its essence aimed at the mechanised production of an object, having as its main techniques prefabrication, transport, and series production.

Although still without great relevance in terms of scale, the interest in modular constructions and prefabricated systems continued to grow, as it was understood that it allowed, above all, for an increase in efficiency, productivity, and profit in the construction sector, through the reduction of skilled labour, waste, energy, and emissions [5,6].

In the scientific community during the 1990s there was a greater interest in the field of prefabrication in the construction sector, as a result of the introduction of recent innovations. Some of these innovations included computer-aided design (the CAD/CAM connection), manufacturing mechanisation, and the robotisation of the construction process. Together, the development of new technologies and materials and the use of computer systems made it possible to learn about more flexible construction systems capable of providing shorter construction times and fewer work accidents, as well as a new capacity for the execution and assembly of structures of a higher quality than had previously occurred [13–16].

Since then, there has been extensive discussion on this topic, first around trying to understand the advantages and disadvantages compared to conventional construction, clarifying why it was not a more evident alternative, and then focussing on more specific factors and seeking to highlight its strengths, such as the fact of improving productivity (due to mass production), the benefits for the construction industry, or through life cycle assessments (LCA) in terms of time, as it allows for a time reduction of about 40% compared to conventional construction, as well as its environmental performance, and the reduction of construction costs (labour, waste production, use of materials) [17].

However, one of the problems that has been reported is that the benefits and challenges of the widespread adoption of prefabrication as an alternative to conventional construction still need to be carefully evaluated [1,18].

The need to write a paper about prefabrication in the construction industry today comes, on the one hand, from the fact that there are no papers in the literature review that address the state of knowledge of this theme in a comprehensive and integrated way, but rather they approach it in a partial way, and on the other hand, because it is understood that a new scenario has emerged, with the need to comply with the objectives of the Paris Agreement and the 2050 decarbonisation goals, cumulatively with the dissemination of the new SARS-CoV virus, which will also have an impact on the construction industry and the even greater role that prefabrication can have in the future.

As prefabrication (and modular construction) is not a new construction concept, it has attracted a wave of interest due to the changes that should occur under the Paris Agreement, as it has the capacity to offer faster construction processes with less environmental impact. Several factors lead one to believe that the renewed interest is here to stay in the markets, primarily due to digitisation. The new digital tools are contributing to the solid maturation of modular project design, ensuring significant savings throughout the process, such as reduced turnaround time, manufacturing, and assembly, and construction cost savings. The construction industry has also been adopting new materials and the development of prefabricated and modular construction systems, focusing now on environmental, energetic, and sustainability issues.

A report prepared by McKinsey & Company [19] suggests that the construction sector and clients themselves are beginning to develop strategies which adopt more industrialised models, giving as an example another study in the report on modern construction methods in the UK, in which 40% of builders claim to be investing in industrial facilities in the framework of prefabrication or plan to do so in the near future.

The most evident benefits of prefabrication compared to conventional construction, identified by several authors, are associated with:

1. Increased Productivity. According to several studies, the industrialisation of construction results in increases in productivity, quality, and sustainability and, consequently, reductions in deadlines, cost, and labour dependence [20–22].

    The productivity of the construction industry benefits from the implementation of production optimisation techniques already tested and demonstrated in other industries, such as: the transfer of tasks to a controlled manufacturing environment—increase in prefabrication; standardisation/repeatability—panelisation and modularisation; robotisation—optimisation of the amount of labour employed in off-site tasks; rationalisation of production chains—the cancellation of intermediate agents optimizes the efficiency of processes and their continuous improvement.

2. Effectiveness in quality control. The benefits in the environmental dimension cannot be realised without integrating the value chain of industry or improving the quality of final products (either by the introduction of new technologies or by specialisation, or by abandoning the approach based on individual projects and replacing this with an approach based on the manufacture of standard products, which in turn requires the development of new materials, products, processes, and possibly industrial units).

3. Reduction of costs (construction costs and the overall cost of a building over its useful life). It is, however, stressed that this should be seen from two perspectives: the lifecycle costs and the impact that prefabrication can have on them; and the cost of the prefabrication and installation investment itself and how this affects the overall cost savings. Nevertheless, it is still evident that there is no history of cost savings among the projects that follow this model. Indeed, one of the main drivers of cost savings comes from economies of scale, and this requires investment in facilities as well as production optimisation. One study identified that companies achieve a rapid and substantial increase in productivity when they start producing around 1000 units.

4. Reduction of work execution deadlines. The optimisation of the project is pivotal to ensure production efficiencies, with mass standardisation and customisation combined with ease of transportation and assembly. If there is a tendency for more time to be needed at an early stage of the construction process, since every design model is already outlined for the execution and manufacturing process, this allows for earlier decision making. It is therefore an advantage over conventional construction because late changes and adaptations are common in the latter, often at a stage when a project is already in the execution/construction phase which will make the process more expensive as a rule. In a second phase, the definition of all elements and components will allow for the development of modular block libraries, which will then make the process more systematised and lower costs. However, the ideal would be, even within the optimisation of the project, to have feasibility for a certain amount of customisation of the models, allowing the client to have some customised features.

5. Greater control of construction time and costs. It allows for a 20%–50% faster construction time than conventional buildings (on-site). The integrated processes involved in prefabrication (including modular construction) are able to eliminate subcontracting costs with on-site labour savings and their associated profit margins in the subcontracting process. With increased repetition of the elements/components or modules, it will be possible to further reduce the associated costs.

6. Automation. With the introduction of these technologies in the manufacturing process being possible, an improvement in productivity will be viable, knowing that this implies a significant initial investment, which then, with successful growth and economies of scale, will be covered by the production costs and respective profit margins.

All of these changes and transformations that are taking place at a global level are leading to measures that re-evaluate the way the built environment is inhabited, and how sustainable development will have to be addressed and ensured for future generations. Regarding the construction sector, the goal of achieving an overall improvement in terms of energy resources and their use is implicit. In this context, the construction carried out in various types of buildings will have to find more appropriate systems and methods to ensure better and more capable energy and resource efficiency.

The construction industry has already been developing other systems which tend to substantially increase productivity, in addition to the conventional construction system, seeking to find innovative solutions that validate a better optimisation of resources and charges and a better life cycle performance, such as prefabricated and modular construction systems.

The literature on prefabrication in the construction industry shows that it is possible to reduce costs and impacts compared to conventional construction; however, most approaches generally analyse the various factors and areas involved partially or for specific contexts [23].

## 2. Prefabrication: New Advances, Challenges, and Opportunities

In a construction process called the "conventional" or also "on-site" method, or even "site-built", a building is built on the site for which it was designed. The prefabrication method is rather an industrialised process based on the production and pre-assembly of off-site components and elements which will subsequently be transported to the site to be assembled, in the form of an open or closed system. In relation to an open system, it can be said that it is the one that allows for the integration of construction materials from different manufacturers, while in a closed system, all components are defined by the system itself or the manufacturer, not allowing for the integration of others. Different types of prefabrication are also considered: the manufacture and sub-assembly of components (e.g., doors), non-volumetric pre-assembly (e.g., wall panels or wooden structures), volumetric pre-assembly (e.g., bathrooms), and modular construction (e.g., complete units or modules that make up a building) [7]. One of the particularities of this system is that substantially, as the level of prefabrication increases, both the use of materials and the energy and greenhouse gas emissions tend to decrease [23]. Prefabrication has been approached as an alternative to conventional construction mainly for residential buildings, but its implementation is still not very expressive [24], despite a growing interest, mainly in the business sector, as it is seen as an opportunity to reduce costs and, above all, environmental and energy impacts.

Overall, most of the existing research addressing comparative studies between conventional construction and prefabrication concluded that prefabricated construction methods are more beneficial, as they have more positive impacts on the environment, the economy, and society. One of the highlights of the literature is its contribution to the sustainable development of this type of system, as it ensures better cost-effectiveness, better control of construction quality, greater worker safety, and consequent labour productivity [11,25–27].

### 2.1. New Advances and Challenges

2.1.1. Sustainability Assessments of Prefabrication Construction

Despite some consensus on the benefits of the prefabrication system, several authors have sought to obtain more practical and concrete data, comparing the various impacts and results of this method with those of the conventional construction method. One of the ways this has been analysed is the impact through lifecycle analysis (the LCA approach), although in reality it has been verified that most of these analyses do not include all phases of the process. This methodology began to be applied in multiple studies in the 1960s, although the first studies on the LCA of buildings were published only 30 years later. In the specific context of the prefabrication method, especially from the year 2000 onwards, the first studies on LCA and the environmental impacts of buildings emerge, although most studies usually focus on one case study and a specific context (for a given location, for a given level

of performance, etc.), which may not allow for an assessment at the level of economy of scale. One of the most recent [5] was carried out with the support of an assessment of the lifecycle of a single-family dwelling, in order to compare the two construction systems, and simultaneously, the use of different common materials. The results showed that the system with a wooden framework (WF) had the lowest impact, followed by the light steel framing system (LSF), and that the incorporated impacts may represent more than half of the total impacts. Compared to conventional systems, prefabricated buildings reduce GHG emissions, energy use, resource scarcity, and damage to both health and the ecosystem. In addition to the reduction of incorporated and operational carbon, prefabrication costs about 30% less than conventional construction. Another conclusion was that material costs accounted for more than 50% of the total cost, although costs in the use phase were not taken into account, and the cost reduction due to large-scale production, which was predicted, was not accounted for. It was also found that construction costs are rather sensitive variables in terms of local costs (e.g., labour and materials), not forgetting the importance of the transport costs that should be considered [5].

In another perspective, various assessments were carried out considering some variations for the same object of study or for some of its elements/components, usually single-family dwellings, in which the influence of several alternatives is analysed, such as different locations (e.g., addressing different climates, transport costs), or different levels of isolation, as a way to identify improvements and ways to reduce the impacts generated. One of these studies assessed a set of these variables for the behaviour of a prefabricated single-family dwelling with one bedroom, in seven different locations (a tropical region, a temperate Mediterranean region, and a continental region) and for three levels of insulation, considering the raw materials used, their transportation to the factory, their factory production, their transportation to the site and assembly, and also the "operational" phase of use (energy consumption by the heat pump, hot water, and other appliances; the use of materials and replacement of refrigerants), and the authors concluded that the data referring to the type of climate and insulation are important to minimize the impacts of the life cycle [28].

Tian and Spatari [29] assessed the environmental life cycle of pre-fabricated residential constructions in China, investigating the uncertainty of parameters on regional greenhouse gases related to electricity production, and concluded that, in general, the prefabrication method offers more environmental advantages compared to traditional methods. Similarly, Mao et al. [28] applied the LCA to determine GHG emissions by comparing a prefabrication system and a conventional construction system in China. The results indicated that the prefabrication system reduced GHG emissions by 3.2% compared to the conventional system. Similarly, Xiao et al. [29] sought to calculate the carbon footprint of a prefabricated component—concrete stairs. The results provided a benchmark for evaluating the carbon emissions of other prefabricated components and developing more sustainable strategies.

The truth is that the LCA, being a "cradle to cradle" approach, must consider all phases of a "product", but in practice, what has been addressed are mainly the environmental and economic impacts by obtaining data on the energy used at different stages of the life cycle, energy consumption, production/assembly costs, and waste. More recently, this type of approach has been a very useful tool to support decision making, as it allows for the attainment of a systematic and quantitative analysis which has been lacking, especially in the past, in the context of an analysis of the impacts of the prefabrication system, or for comparative terms [30]. Evidently, this type of approach requires the use of more specific software, such as building information models (BIM) integrated with energy simulation software and with data from inventories and a life cycle model, as well as sensitivity analyses. To this end, several authors [31] have used complex evaluation models, such as the cloud model, performance evaluation methods, and other methods for determining parameters affecting the comprehensive benefits of prefabricated buildings, such as the study by Zhou et al. [32].

### 2.1.2. Models That Incorporate Specific Software

Some authors, as a way to dispel some gaps in the analysis of the impacts of the prefabrication system, have tried to develop digital models, supported for example in BIM, as a way to differ from most methods, which are generally based on criteria assessments according to specific objectives. One of these studies sought to propose a construction method based on a set of criteria that assess economic, social, and environmental considerations, as this is one of the gaps highlighted in other articles (usually partial assessments of impacts) with a special focus on the environmental aspect. In this way, it would be possible to analyse the system so as to achieve construction sustainability. The available information was compiled using a digital platform (BIM) integrated with automated algorithms to quantify the criteria applied. The evaluation criteria are defined for each economic, social, and environmental impact, in which two factors were considered for each sustainability parameter; for example, for the environmental scope, the factors reported on are emissions and energy, or in the case of the social scope, the factors considered are noise pollution and the reinforcement of worker safety. Although there are limitations resulting from the limited number of evaluation parameters, this type of model, if developed by integrating systematised algorithms and various factors for each parameter in accordance with the type of building and other construction solutions, can constitute a digital tool with potential for the evaluation of the most appropriate prefabrication system for each type of building [12].

In the same context, Mohammad Kamali et al. [33] proposed to assess the life cycle sustainability of different construction methods (conventional vs. prefabrication) by including sustainability performance indicators (SPIs), where once again, the crucial point lies in selecting the appropriate SPIs based on a severity index. The study in question applied surveys of professionals in the sector as its methods, and the results show an increase in the social impact compared to the environmental impact in the evolution of the sustainability of construction. Liu et al. [31] developed an index as a method of assessing the safety performance of prefabricated framing systems using a computer method based on this type of model. Other methods have been reported, such as the development of a building inventory model, demonstrated as a reliable way to predict market dynamics when introducing technological innovation.

In another study [34], two prefabrication systems were analysed: (1) a light steel and wood frame with OSB panel walls; (2) a conventional system with a concrete structure and brick masonry. The impacts and costs were analysed for several EU cities, and the results were calculated for each country. The results showed that the prefabrication method can reduce the impact and costs of building stock compared to conventional systems. For example, the former can be more easily dismantled and their materials recycled, which leads to lower end-of-life impacts (EoL), less waste, and higher rates of reuse and recycling. It is also possible to more easily reduce the materials and labour used and the construction time. The prefabrication method also allows for construction waste to be substantially reduced from 10–15% to less than 5% compared to the conventional method with larger possibilities for recycling of the waste in the production/manufacturing phases [35].

System Dynamics (SD) was another method analysed by some authors [1,36] in order to respond to the need for the management and control of the prefabrication system, seeking a cost optimisation and analysis of the benefits. This type of approach provides methods and tools for modelling and analysing system dynamics. The results of the models can be used to communicate essential results to help everyone understand a system's behaviour.

### 2.1.3. Analysis of the Main Impacts

According to professionals in the construction industry, economic impacts still play the most significant role in differentiating sustainability between the two construction methods, and this will lead to the decision being taken for a gradual commitment to the prefabrication method, once some limitations of the criteria involved can be solved compared to the conventional method.

According to the European Construction Sector Observatory, based on Eurostat data, the total number of people employed in the construction sector in the EU-28 reached 21.1 million in 2015. The same observatory also reveals that two out of five companies involved in construction claim to have difficulty in recruiting workers with the necessary skills for existing vacancies [37].

Overall, this trend reflects the fact that the mismatch between labour supply and demand in the market is increasing, making it increasingly difficult for companies to find employees who have the skills needed for existing vacancies. Similarly, one of the concerns of the construction sector is its labour productivity. In addition to low labour productivity, the construction sector has also to deal with exponentially increasing construction costs, mostly as a consequence of the recent environment. As one can see in Figure 1, among all construction costs, the cost of labour was the one that increased the most.

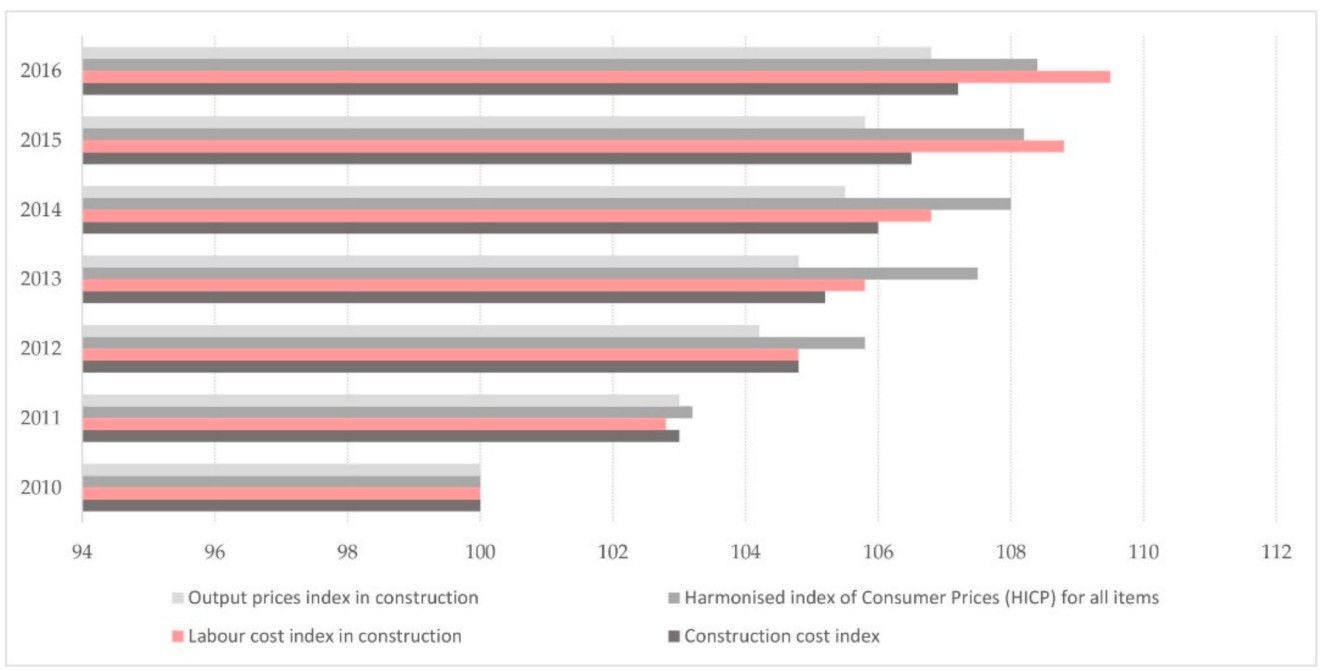

**Figure 1.** Construction costs index in the EU-28 between 2010 and 2017 (2010 = 100).

Indeed, it is understood that investing in prefabrication will ensure greater rigour and control of economic aspects through not only the reduction of general costs and impacts, but also, in the case of labour control, through the qualification of workers (skilled labour) and the improvement of safety and working conditions (greater control and reduction of risks).

From research on the environmental impacts of prefabricated buildings, the main results indicated that one of the greatest advantages of prefabrication is the positive impact on improving the rate of resources use, indirectly contributing to a reduction in construction waste. Very recent data also indicate that the useful life of a prefabricated building can be up to 100 years, and the useful life of its components can be up to 20–30 years, although these results are from a study in a specific context [1]. Questions about both construction methods with regard to the environmental impacts are closely related to assessments of embedded energy (EE) and greenhouse gas (GHG) emissions. A study on a prefabricated house found

that the incorporated impacts increase with the size of the house. On the other hand, transport impacts (of modules, workers, and finishing materials) vary significantly depending on their location [38]. Although most of the studies carried out show that the prefabrication method is a promising solution for the construction sector, as it allows for a reduction of costs and impacts, mainly environmental and energy, (which were the most analysed topics), the impact of costs associated with transport may compromise overall reductions and should be further studied to assess the impacts related to transport in prefabricated construction. The most relevant of these will be discussed in this document [39].

As has been reaffirmed in this document, most studies on the prefabrication method mainly address the environmental impacts through an assessment of embedded energy (EE) and greenhouse gas (GHG) emissions. Vanessa et al. [38] addressed a case study of a (modular) house with a high degree of prefabrication using an analysis of several scenarios for which the primary data for manufacturing were collected, including alternative structural materials, house dimensions, and different locations in the EU (a total of seven). For this purpose, a cradle-to-site model was considered, taking five phases into account from materials production to on-site modules assembly and finishing. This prototype was built in two phases: factory production of the modules and transport to the site and assembly. A database developed for construction materials was used for the calculation of the embedded energy and GHG emissions. The different scenarios included different structural materials, dimensions of the house (from one to four bedrooms), and locations. Transport incorporated two main phases: transportation to the factory (of workers and materials) and transportation from the factory to the site (of modules, workers, and materials). For the main results obtained on the impacts of transport in the country of origin, it was first necessary to consider the transport of six workers to the factory in three passenger cars for a distance of 10 km. Then, the transportation of materials required a truck to drive a distance of 50 km, and for transportation from the factory to the site, each module was transported individually in a truck. In the baseline scenario in the country of origin, transport represented 2% of the total impact. In the case of a more distant location, in another country, transport to the site may represent a significant part of the total impact, reaching 25% of EE and 27% of GHG. Another conclusion on the impact of transport to the site was that it does not increase linearly with distance but depends mainly on the mode of transport. Transportation is an issue with considerable impact since it is a constraint that may limit some variables of the development and construction process, such as the size and weight of the modules, distance, mode, and transport routes [7]. From the results of these studies, it was also possible to conclude that the calculated impacts (cradle-to-site) show that the production of materials is the most important contribution representing between 64 and 90% of EE and 59 and 87% of GHG. The second most important contribution is transport. In order to improve the environmental performance of these systems, it is recommended that special attention be given to the selection of materials with lower energy and carbon consumption and to the reduction of the impact of transport by reducing the distance from the factory to the site; choosing of modes of transport with lower energy consumption; and selecting local materials and labour to optimize assembly on site.

Undoubtedly, the impacts associated with transportation costs can be significant, as prefabrication systems require an extra initial phase in relation to transportation from the factory to the site, loading and unloading, and also in terms of safety, which can represent up to 20% of the total incorporated impacts. Specifically for modular construction this may be a critical issue, since it may limit the size and weight of the modules, and the mode of transport [22,40,41]. Other possible restrictions of the prefabrication method that still need to be improved before and after the design of the structure are the dimensions of the modules and the inability to make changes on site [42].

### 2.1.4. Opportunities

The transformation process of the construction industry has occurred more or less gradually, but in recent decades with the need for a construction industry based on the pillars of sustainability, other methods have been developed, such as prefabrication, in which the aggravating factor of the pandemic crisis has contributed to speed up some changes in the ongoing process. Initially, the greatest impact of the pandemic crisis was due to the strict restrictions on the mobility of people and, in the work context, of companies. The economic and social impact was the most impactful, with one of the main post-pandemic effects being an even deeper economic recession than the 2008-2009 global financial crisis or the Latin American debt crisis of the 1980s, but so far it is not possible to draw firm conclusions [3].

Overall, the main consequences of the pandemic crisis were social isolation, a reduction in production and supply of the markets, insolvencies/redundancies in companies with a consequent increase in unemployment, a reduction in global growth that limits consumption, investment, and exports, as well as many uncertainties about the future. One of the sectors affected was the construction industry, mainly due to the lack of raw material and the significant increase in costs, although despite early negative expectations, the sector is currently suffering less than expected and recovering quite quickly. However, in this current scenario of some optimism there are still concerns for the future about the long-term impacts of the crisis. However, some positive consequences have been pointed out, in the sense that there is a greater investment in digitalisation, an acceleration in the manufacturing industry, a greater investment in construction systems that are alternatives to the conventional systems—ones that are more sustainable and faster, such as prefabrication—which represents a strong investment in innovation by companies, and also an investment in the standardisation of building codes and the manufacturing process. There will tend to be a need for a faster response in terms of work execution times, the systematisation of the process, and the safety and health of workers, with the aim of being able to face global crises and bad weather, often as a result of climate change, better and faster. It should also be noted that the risks of delays due to extreme weather conditions, vandalism, and site theft can be minimised with prefabrication systems.

In the coming years, this fundamental change is likely to be catalysed by changes in market characteristics, such as a shortage of skilled labour, the pressure on infrastructure and material costs, stricter legislation on sustainability and safety/health, further industrialisation and the emergence of new materials, the digitisation of products and processes, and new players/actors and new demands by the client/owner, which will shape the future dynamics of the construction industry.

### 3. Discussion

The main focus of this document is to outline the potential of prefabrication in the construction industry, based on comparative data with the conventional method, whose growing interest has been due to the goals established for 2050, climate change, and the consequences of the impact of the spread of the new SARS-CoV virus. The existing literature has generally considered that this method contributes to an effective reduction in the overall impacts and costs of building and the harmful consequences for the environment. Through the literature review, it was possible to verify that there were several approaches to establish a comparison between the prefabrication method and the conventional method, as well as its main benefits in order to identify the reasons for a more gradual use. Overall, most existing research addressing comparative studies concluded that prefabrication methods are more beneficial, as they have more positive impacts on the environment, the economy, and society, i.e., they contribute more decisively to the sustainable development of construction.

*3.1. Research Based on Quantitative and Qualitative Data*

Through the different types of approach—quantitative and qualitative—which resulted from the use of various methods and models, presented in the previous chapter, and whose main objective was to compare the various impacts and results of both construction methods, it was possible to identify the most used methods and models for this purpose:

- Life cycle assessment (LCA) [29,43–46] refers generally to an environmental assessment, where the life cycle cost (LCC) and the social life cycle assessment (SLCA) are counterparts to the economic and social assessments, respectively (most commonly used terms: life cycle assessment (LCA); LCA approach; life cycle performance). As a methodology, its structure is standardised by ISO 14040, Life Cycle Assessment, Principles and Framework (2006) and ISO 14044, Life Cycle Assessment, Requirements and Guidelines (2006). This approach represents the great advantage of obtaining important results on the (environmental) impacts of a product, being a very useful tool for decision making in the evaluation of products and processes [24,47].

  The main issue in its application is that as a methodology for assessing and establishing the terms of comparison between two methods, the studies mainly refer to the environmental impacts of buildings the validations of which address specific cases. In addition, the process itself is somewhat complex, mainly due to the difficulty in obtaining some of the necessary data, such as the specifications for various types of materials and components, assembly methods, and end-of-life disposal [24].

  Another aspect identified is the lack of research studies on the economic element, but above all on the social dimension whose life cycle assessment is pivotal for the development of sustainable construction. For example, economic impacts are mostly calculated using LCC analysis which does not capture their effect on economic sustainability, especially when addressing the social dimension whose indicators still need to be further explored, largely because of the difficulty of quantifying some data and their subjective nature. Some of the data verified are the impacts on human health, accidents and safety, public welfare, and equity.

  Decision making by the construction industry regarding the different construction methods could be eased by the combination of life cycle assessment (LCA), life cycle costing (LCC), and social LCA (SLCA) translated into a life cycle sustainability assessment (LCSA), with the goal of obtaining more sustainable products throughout the life cycle. It would be important to integrate MRIO (multi-region input–output) and system dynamics (SD) modelling databases, along with quantitative social and economic indicators [44].

- Models that incorporate specific software [48–52] generally refer to database software in digital format, such as the Building Information Model (BIM), value engineering (VE), or dynamic system simulations (DS). Considering that the optimisation of the built environment can drastically reduce the energy consumption of buildings, this database software is used for optimisation considering both energy savings and life cycle costs. A recent study proposed a theoretical optimisation system to analyse the "green" or environmental credentials of a building based on BIM-VE [53]. Another developed an action-based approach that combines BIM-LCA integration and statistical distributions to better understand the impacts and costs of buildings [34]. System dynamics (SD) is a simulation method that focuses on system interactions over time, and which considers feedback effects. It is suitable for use with energy and material information flows, such as for cost optimisation, inventories, or benefit analyses. Some authors have resorted to the use of SD in the context of the prefabrication method as a way to help identify and quantify the relationships of the relevant parameters of the (sub)systems [1]. The use of these models and methods, or their combination, will allow for more complex designs, thereby improving communication and coordination between stakeholders and ensuring that construction is of higher quality [25]. The technological advances under way should allow for the development of 4D (time) and 5D (cost) modelling, which will result in greater cost reduction and increased efficiency.

3D printing could also be used for prefabrication, allowing for further reduction of any associated human failures.

The main issues are with its application. The existing literature on BIM applied to the prefabrication method allows us to conclude that the use of this tool is normally applied to a specific characteristic and at a certain stage of the construction process. Most of the research addresses case studies using this tool, such as BIM for the design of architectural elements, and the consequent productivity gain (time-cost ratio), or its added value in determining the assembly sequence and confirmation of connections during execution [54]. Other studies are more focused on specific aspects, with the fire fighting and structural modelling HVAC in the scope of prefabrication, or the measurement of carbon emissions reduction during the execution phase of the project [50].

### 3.2. Research Based on Economic, Environmental, and Social Impacts

Using the existing literature reviewed in the previous chapter, it was possible to define the main economic, environmental, and social aspects that result in benefits of the prefabrication construction method compared to the conventional method [47,55,56]. For the economic aspects, the indicators assigned were mainly time and costs. For the environmental aspects—which include the entire life cycle, from the extraction and manufacture of materials, transportation, construction, maintenance, to the end of life—the most referenced indicators are water consumption, $CO_2$ emissions, energy consumption, and solid waste [47]. Regarding the social aspects, one may conclude that this is the least developed aspect in the evaluation of the prefabrication process; however, the main indicators that have been referenced include the safety of workers and occupants (work-related risks during construction and assembly, and risk of accidents, ease of expansion) and adaptability to changes (ability to assemble/disassemble and replace) [47,57,58].

In most of the different aspects and their indicators, it was possible to conclude what motivates the growing interest in the prefabrication method, mainly:

- Increased productivity (e.g., with increased automation) [59–62];
- Prefabrication construction has a higher initial cost, but with shorter deadlines and smaller economic and temporal deviations [9];
- Reduction of construction costs [60,62,63];
- Greater control of construction time (deadlines) [62,63];
- Decrease in labour costs and the amount of on-site labour [59,60,63];
- Prefabrication construction allows for a lower environmental impact in the construction and end-of-life phases, during which less energy and water is consumed and emissions are reduced (greater control and optimisation of the construction process and of a building's useful life) [29,59];
- Reduction of the negative impact on the environment (reduction of consumption and emissions) [29,59,64];
- Reduction and greater control of construction execution times and of a building's useful life (from production to the end of life) [59,64];
- Greater efficiency in quality control and fewer failures due to the degree of accuracy (since there is greater control and rigour in the initial phases due to standardisation and factory supervision) [29,59,61,65];
- Reduction of construction waste and scrap (optimisation and control of the manufacturing process) [29,47,59,65];
- Reduction in the risk of work accidents (most factory work allows for a more controlled environment with less risk exposure) [60,66];
- Greater control of the safety and health of workers [60,61,66].

However, some authors have acknowledged several issues that may still restrict decision making towards the added value of the prefabrication construction method, which are essentially: impossibility or great difficulties associated with changes to the construction site; limitations of architectural solutions or transport restrictions (such as dimensions of materials, load capacity, and travel and transportation costs).

In practice, the implementation of the prefabrication method requires a higher initial cost, but it is able to guarantee shorter deadlines and fewer economic and time deviations. The main benefits have already been listed, of which the environmental impact is the most prominent, since the various studies carried out indicate that the prefabrication method allows for a lower consumption of energy and water and a reduction of emissions. It is a construction method which takes place under optimised conditions subject to greater rigour and control, from the initial stages and throughout the construction process (production, logistics, and assembly), and then at the end of life (demolition, deconstruction), where most of the elements/components can be dismantled allowing for their reuse and recycling. Effectively, the research into ways to minimise the impact of transport should be further developed.

In general, the existing literature considers that, in comparative terms, the prefabrication method has numerous advantages over the conventional one, although both issues and benefits have been recognised. However, in addition to representing a higher initial investment cost, two aspects should be noted: one that should still be improved is the aesthetic/architectural dimension of prefabricated buildings, since it was verified that the client/occupant is aware that it is not possible to obtain the same type of innovative solutions and materials at an organisational and aesthetic level compared to the conventional construction method; the other refers to the impacts associated with transport. Several authors who address this aspect agree that the prefabrication method requires extra costs in relation to the transportation of materials from the factory to the site, loading and unloading, and also in terms of safety, which can represent up to 20% of the total incorporated impacts [7]. Consequently, this construction method will have to develop ways to overcome some critical difficulties related to dimensional limitations and weights of the modules, modes of transport, and possible delivery delays, especially for long distances.

## 4. Conclusions, Recommendations, and Future Perspectives

Recent events, the effects of climate change, and the pandemic crisis have brought an increased degree of uncertainty about the future path of the construction industry, and the various associated implications underline the need to rethink construction methods and their sustainable development in terms of greater rigour and control of their costs and impacts.

This entry allows for the clarification of the new advances, challenges, and opportunities of the prefabrication method, noting that most studies are based on comparisons between this and the conventional construction method to carry out analyses of concrete aspects, and case studies to draw conclusions. The main findings on the impacts of the prefabrication method on the construction industry were discussed in Section 4, through the correlations between the diverse existing literature, the main results obtained by the various authors, and the issues identified.

However, in light of the constraints resulting from recent events, it is also expected that the impact of the COVID-19 pandemic will bring a long-term dynamic that is still difficult to foresee. However, some important aspects can already be seen, such as the shortage of specialised labour (an important complication in several markets), in addition to the lack of good working conditions to attract and retain the required workforce, many of the remaining members of which are reaching retirement age. Another important aspect relates to issues of sustainability and safety requirements in the workplace, which have been further accentuated by the dynamics of the consequences of climate change and increased pressure on the construction industry to reduce carbon emissions. At the same time, some markets are recognising the need for regulatory standardisation on sustainability and

safety and building codes. As the prefabrication and automation of off-site production allows for a more controlled and rigorous construction approach of the entire life cycle, the industrialisation process is taking an increased interest in significant improvements at various levels for all involved in the construction process.

There is also a greater interest in the innovation of new materials and their application that allow for a reduction in carbon footprint, as well as a growing commitment by the business sector to digitise products and processes through digital technologies. These will allow for greater control of the entire value chain and a shift to more data-driven decision making, ensuring more efficient operations, as well as new business models, such as the digital channels that are opening up for construction, with the potential to transform the interactions of buying and selling goods through the value chain.

The current context and the medium and long-term repercussions are causing changes in the construction industry. One of these changes will be a product-based approach in which quality is considered a measurable and accurate set of characteristics required to satisfy the consumer, such as defining the characteristics of the product according to the expiration date/warranty of a product. Increasing specialisation needs and investments in innovation—including the use of new materials, digitisation, technology, and installations—are also expected. Companies will tend to focus on niche specialisation and target segments as a way to improve their margins and differentiation levels, and this will tend to control potential cyclical risks or benefits, thereby ensuring a construction industry with a significant level of consolidation. For companies, the prefabrication method will require greater investments, for example, in facilities, manufacturing equipment (manufacturing automation), in technology and specialised human resources, and in research and development. Innovation, digitisation, value chain control, the use of technology, and specialisation increase the importance of development and more specialised knowledge, but ensure an overall reduction of costs and impacts; greater control of sustainability, productivity, quality, and rigour; and a decrease in the degree of uncertainty in the construction industry.

**Author Contributions:** Conceptualisation and editing, P.F.R.; Writing—review, N.O.F. and F.P.; Supervision, N.B.P. All authors have read and agreed to the published version of the manuscript.

**Funding:** This research was funded by Agência Nacional de Inovação (NORTE-01-0247-FEDER-047054). The APC was funded by Agência Nacional de Inovação (NORTE-01-0247-FEDER-047054). The funding sponsors had no role in the design of the study; in the collection, analyses, or interpretation of data; in the writing of the manuscript, and in the decision to publish the results.

**Institutional Review Board Statement:** Not applicable.

**Informed Consent Statement:** Not applicable.

**Data Availability Statement:** Not applicable.

**Conflicts of Interest:** N.B.P. is the president and founder of CICON, also partner and employee of Houselab, N.O.F. is the partner and employee of Houselab, F.P. is the partner and employee of Houselab, P.F.R. is an employee of Houselab. The authors declare that they have no known competing interests or personal relationships that could have appeared to influence the work reported in this paper.

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
