# Peer review of "Impacts of Prefabrication in the Building Construction Industry"

_encyclopedia, doi:10.3390/encyclopedia3010003_

Round 1

Reviewer 1 Report

This paper presents a literature review to gather relevant information on the new advances, challenges, and opportunities of prefabrication in the building construction industry. This topic is interesting and also deserves to be investigated. Here are the main comments for consideration.

1. Please highlight the necessity and contribution of this paper in Introduction part.

2. State of the art needs improvement. Please avoid lumped citation, e.g. [12-15]. A detailed description of the cited references is essential. Several recently published papers are not included in the review section. In fact, the acknowledgment of the past related work by others, in the reference list, is not sufficient. Kindly note that references cited must be up to date.  

3. Please thoroughly check the manuscript. Some typos and unclear sentences can be found, e.g., the subsection of 2.1.2 is not mentioned.

4. Please recheck all of the citations used in this paper. Some mistakes can be found, e.g., in line 349, Yimeng et al. [29] should be rewritten into Tian and Spatari [29].

Author Response

  1. Please highlight the necessity and contribution of this paper in Introduction part.

REPLY:

At the end of subsection 1.1 we had indicated the necessity and contribution of this paper. We then move on to the end of section 1- Introduction.

“This document presents a comprehensive and integrated review on the various scenarios of prefabrication development in the construction industry compared to conventional construction. Also, with the aim of outlining the potential of prefabrication in the construction industry, in the current context and in light of the goals set forth until 2050, and the consequences of the impact of the spread of the new SARS-COV-2 virus in this market, the review will also include the approach of this conjuncture, seeking guidelines for the future.

One of the strong points of this document is that it aims to aggregate a broad set of bibliographies on this topic, which is somewhat dispersed, identifying gaps and opportunities.”

  1. State of the art needs improvement. Please avoid lumped citation, e.g. [12-15]. A detailed description of the cited references is essential. Several recently published papers are not included in the review section. In fact, the acknowledgment of the past related work by others, in the reference list, is not sufficient. Kindly note that references cited must be up to date.

REPLY:  Ok. We consider the reviewer's suggestion to check other recently published articles.

  1. Please thoroughly check the manuscript. Some typos and unclear sentences can be found, e.g., the subsection of 2.1.2 is not mentioned.

REPLY:  Subsection 2.1.2 is mentioned in the text.

  1. Please recheck all of the citations used in this paper. Some mistakes can be found, e.g., in line 349, Yimeng et al. [29] should be rewritten into Tian and Spatari [29].

REPLY: OK. We have made the correction.

Reviewer 2 Report

The level of contribution is low. There are many review articles about prefabrication. This does not provide any new information.

The authors stated that "there are no papers on the literature review that address the state of knowledge of this theme in a comprehensive and integrated way." However, there are a lot of prefabrication review articles in the Scopus database. 

The research article is not just to write the essay and then put all references to the end of the paragraphs. 

Author Response

Dear Reviewer,

There are indeed many articles about prefabrication in the Scopus database, but as mentioned in the abstrat of the paper:

“Therefore, this entry seeks to review the existing literature on prefabrication, seeking to gather relevant information on the new advances, challenges, and opportunities of this construction method whose approach has been mostly focused on partial or specific aspects for case studies, both highlighting the potential and identifying gaps and opportunities of prefabrication in this new context”

It's also mentioned that:

This document presents a comprehensive and integrated review on the various scenarios of prefabrication development in the construction industry compared to conventional construction. Also, with the aim of outlining the potential of prefabrication in the construction industry, in the current context and in light of the goals set forth until 2050, and the consequences of the impact of the spread of the new SARS-COV-2 virus in this market, the review will also include the approach of this conjuncture, seeking guidelines for the future.

One of the strong points of this document is that it aims to aggregate a broad set of bibliographies on this topic, which is somewhat dispersed, identifying gaps and opportunities.”

Round 2

Reviewer 1 Report

All the comments have bee fully addressed.